# Effects of Multistrain Probiotic Supplementation on Sows’ Emotional and Cognitive States and Progeny Welfare

**DOI:** 10.3390/ani14060847

**Published:** 2024-03-09

**Authors:** Melody Martins Cavalcante Pereira, Ines Andretta, Carolina Haubert Franceschi, Marcos Kipper, Alexandre Mariani, Thais Stefanello, Camila Carvalho, Júlio Vieira, Luiene Moura Rocha, Andrea Machado Leal Ribeiro

**Affiliations:** 1Department of Animal Science, Universidade Federal do Rio Grande do Sul, Porto Alegre 90010-150, Rio Grande do Sul, Brazil; melody.zootecnia@hotmail.com (M.M.C.P.); carolfranceschi3@hotmail.com (C.H.F.); alexandre.bonadiman.abm@gmail.com (A.M.); thaisstefanello@gmail.com (T.S.); vieirajulio16@gmail.com (J.V.); aribeiro@ufrgs.br (A.M.L.R.); 2Elanco Animal Health, São Paulo 04794-000, São Paulo, Brazil; marcos.kipper@elancoah.com; 3Department of Pathobiology, Pharmacology and Zoological Medicine, Faculty of Veterinary Medicine, Ghent University, 9000 Gent, Belgium; camila.lps.carvalho@gmail.com; 4Quebec Pork Development Center, Deschambault, QC G0A 1H0, Canada; luiene.moura@gmail.com

**Keywords:** microbiota metabolites, feed additive, gut–brain axis, piglets

## Abstract

**Simple Summary:**

Feeding technologies, such as probiotics, are widely used in the animal industry for purposes beyond nutritional needs. Probiotic bacteria have been shown to have multiple effects on sow physiology during gestation and lactation, making them a subject of great interest when it comes to animal behavior and welfare. In this study, the use of probiotics mitigated the apathetic state of gestating sows housed in individual cages, in addition to reducing fear and aversion towards humans during interaction tests. The influence was also observed in the progeny, as piglets from sows supplemented with probiotics were less resistant in fear assessments, which may indicate lower stress levels. The findings indicate that the use of probiotics during gestation and lactation can be used as a tool in enhancing animal welfare.

**Abstract:**

The intensification of production systems has resulted in detrimental effects on sow welfare, which can have an adverse influence on their offspring. Considering the relevance of the microbiota–gut–brain axis, probiotics can mitigate such impacts. To investigate the effects of the dietary inclusion of probiotics on the welfare of sows and piglets, 147 multiparous sows were randomly assigned to two groups: a control group or a group supplemented with a multistrain probiotic from the beginning of pregnancy to the end of lactation. The human–animal relationship (HAR), stereotypic behavior, position changes, salivary cortisol, and plasma serotonin levels were assessed in the sows. The piglets’ back test behavior and organ weight were analyzed. The probiotic-supplemented sows exhibited a better HAR index (*p* = 0.017), which indicated reduced aversion towards humans. The frequency of stereotypies was not influenced by the treatments. However, the supplemented sows spent more time standing (*p* = 0.054) and less time lying down (*p* = 0.008). The cortisol level of the supplemented sows was 50% lower (*p* = 0.047) and the serotonin levels were 11% higher (*p* = 0.034) than control animals. The multistrain piglets were more passive and less resistant (*p* = 0.076) in the back test. The organ weights were not influenced by treatments. In conclusion, the sows supplemented with probiotics showed less fear and more motivation indicators, while their piglets showed less aggression behaviors.

## 1. Introduction

Intensive production systems were developed to improve productive indices. and their ongoing progress is noticed year after year. However, it is essential to acknowledge that some intrinsic factors of these production models can potentially lead to negative impacts on animal welfare and subsequently influence animal physiology [1,2].

Various studies have consistently evidenced the beneficial impact of incorporating probiotics in the diets of sows on the overall performance of the animals. These effects include enhancing piglet birth weight, the quality and quantity of colostrum, and feed consumption during lactation, with positive consequences for body score at weaning [3,4]. Recently, the hypothesis that probiotic bacteria are also a promising strategy to mitigate the adverse effects on animal welfare caused by intensive farming practices has garnered significant interest within the industry.

Research has provided evidence that feed additives capable of modulating the intestinal microbiota can produce positive effects on the expression of stereotypies [5,6,7]. This type of behavioral pattern is associated with the occurrence of apathy and lack of motivation and may cause negative results in social interactions among sows and between sows and handlers. These effects not only increase the occurrence of injuries to those involved but exacerbate the already established stress [8].

The microbiota–gut–brain axis is already well established in other research areas as an intersection of microbiology and neuroscience. This concept relies on the influence that intestinal bacteria have on neural processes, the activation of the hypothalamus–pituitary axis (HPA), and behavior expressions [9]. These communication networks are complex, involving the exchange of immunological, neuronal, and chemical signals [10]. However, the bidirectional communication between gut bacteria and the brain certainly plays a critical role in the current understanding of animal behavior. Therefore, lactic acid bacteria can alter an individual’s behavior by affecting the brain through the regulation of hormonal (such as serotonin, dopamine, cortisol, and more) patterns via the HPA [11]; by inducing anti-inflammatory cytokines and reducing proinflammatory cytokines [12]; by regulating the pattern of neurotransmitters in the body; and by the production of metabolites such as short-chain fatty acids [13].

Notably, the use of probiotics belonging to the *Lactobacillus* strains have shown effects on the reduction in behaviors associated with anxiety and depression in mice [9,14]. In addition to the direct effects in supplemented animals, probiotics administered to sows may confer advantages to their offspring as well. Previous studies have already proved the great maternal influence on the first microbial colonization of the piglets [15]. Complementary, certain behavioral conditions can also be transferred from mothers to offspring. For instance, more stressed and aggressive sows can consequently produce more stressed and aggressive piglets [16,17]. Considering the relevance of the topic for human and animal sciences, this study was carried out to evaluate the impact of including a multistrain probiotic additive in the diets of gestating and lactating sows on various aspects of behavior and welfare of both the sows and their litters.

## 2. Materials and Methods

The experimental protocol was approved by the institutional Animal Care and Use Committee of Universidade Federal do Rio Grande do Sul (register number: 39736).

### 2.1. Farm, Animals, Housing, and Management

This study was conducted on a commercial farm located in the Brazilian state of Rio Grande do Sul, which experiences a subtropical climate.

A total of 147 sows (Pic Camborough, Agroceres-PIC, São Paulo, Brazil) were used in the trial. The initial number was 166, but some animals were removed during the study due to mortality or reproductive failures. Animals with parity orders from 2 to 9 were used in the trial but were equally distributed within each treatment.

The gestating and lactating sows were housed in different facilities, which were naturally ventilated, and an evaporative cooling system was activated on the gestation house when necessary. The gestation housing consisted of standard individual crates (1.2 m^2^) equipped with automatic feeding and drinking systems. During pregnancy, temperature control was performed by using curtains and fans. The farrowing housing consisted of a crate (1.2 m^2^) for each sow complemented with a circulation space (4.2 m^2^) plus a creep-heated area (1 m^2^) for the piglets. Manual feeders and nipple drinkers were used. During lactation, the temperature was controlled by using curtains and an air conduction heating system with a thermostat, and for piglets, floor heating and incandescent lamps were used.

During the study, almost all the existing animal husbandry practices on the farm were upheld, including feeding practices, farrowing attendance, and weaning ages. These practices are representative of modern and intensive pig production in Brazil. However, some extra activities were implemented to guarantee the scientific quality of the study. For instance, the accuracy of the automatic feeders was checked weekly in the gestation phase, and the amount of feed offered during the lactation period was quantified. In addition, when implementing the cross-fostering management (already practiced in the farm to equalize litter size), extra attention was undertaken to ensure that the animals were not transferred outside their designated treatment.

### 2.2. Experimental Design, Treatments, and Feeding Practices

Each sow/piglet was considered an experimental unit in this study. The number of replicates used in each test was defined for each response considering the data dispersion observed in previous trials and using the tools available in the Minitab software (v. 20, Minitab Inc., State College, PA, USA).

The treatments were randomly assigned to the experimental units within each block (parity order) in such a way that each treatment had the same number of animals of each parity order. The final arrangement led to a parity order distribution that was very similar between the treatments (mean = 4.45 × 4.39; mode = 4 in both groups; first quartile = 3 in both groups; and last quartile = 6 in both groups).

Two treatments were evaluated: (1) a control, in which the animals did not receive any probiotics; and (2) a probiotic group, in which the sows received a probiotic (Protexin™ Concentrated, Elanco™ Animal Health, São Paulo, Brazil) with *Lactobacillus acidophilus* (2.06 × 10^8^ CFU/g), *Lactobacillus bulgaricus* (2.06 × 10^8^ CFU/g), *Lactobacillus plantarum* (1.26 × 10^8^ CFU/g), *Lactobacillus rhamnosus* (2.06 × 10^8^ CFU/g), *Bifidobacterium bifidum* (2.00 × 10^8^ CFU/g), *Enterococcus faecium* (6.46 × 10^8^ CFU/g), and *Streptococcus thermophilus* (4.10 × 10^8^ CFU/g). The supplementation was provided only to the sows, lasting from right after the artificial insemination until the weaning of the piglets at 21 days of age. The piglets were not supplemented. To prevent cross-contamination (among sows, from the feed to the piglets, and in the environment), pharmaceutical gelatin capsules were filled up with the specified amount of the probiotic to each sow on each day. These capsules were then provided to each sow during the trial. The capsules were placed on the top of the feed served to each sow in the morning. Red capsules were used to allow easy observation of its intake by the sow. It was verified in all cases that the animals consumed the capsule.

A feed based on corn and soybean meal was used in both phases. Formulation was performed following conventional nutritional levels for Brazilian pig production [18]. During the gestation phase, 1.9 kg of feed was provided once a day (morning) per sow. Adjustments of this amount were performed for sows with inadequate body condition scores, with the daily feed intake varying from 1.8 to 2.0 kg. During the lactation phase, the amount was supplied ad libitum, with the daily feed intake gradually increasing during the first week, reaching an average of from 7 to 8 kg per animal. Despite being supplied in the gelatin capsules, the daily amount was weighed individually to represent an inclusion of 50 g of the additive per ton of feed, which is the concentration recommended by the supplier (Elanco™ Animal Health, São Paulo, Brazil), which resulted in a mean daily supply of 0.1 g during gestation and 0.4 g during lactation per sow.

### 2.3. Sow Assessments

The repertory of stereotypic behaviors presented in the farm was built in a preliminary study that lasted for a week. The following behaviors were registered: sham chewing, tongue rolling, bar biting, and floor licking. The test was conducted out in the mornings (one hour after feeding) during the final week of gestation (before transfer to farrowing facilities). Each sow was observed for three minutes, with the first minute disregarded. A score of 0 was assigned when no stereotyped behaviors were observed. A score of 1 was attributed if any stereotyped behavior was observed during the observation time. To disregard the “novelty” factor, a single evaluator was used, always wearing the same clothes. During the observation time, the evaluator noted on a specific form whether stereotypies occurred.

The body posture was also observed and registered during the same observation period (adapted from [19]). Sows were classified as standing, lying, lean forward, and sitting. Changes in posture during the period were also registered. In this regard, a score of 0 was attributed to no change in posture, while 1 was attributed when a change in posture was observed.

The human–animal relationship (HAR) observational test was also conducted out in the mornings and afternoons during the final week of gestation (before transfer to farrowing facilities), right after the previously described test. A simulation was performed in the week before so that the sows could get used to the analysis and the novelty factor would not affect the animal response and to understand some behavior patterns, such as grunts. The test consisted of three distinct steps. In the first step, the evaluator approached the sow from the front, positioning himself slightly to the right of the sow for 10 s. After, the evaluator crouched down in front of the sow and remained motionless for 10 s. Lastly, the evaluator reached out and attempt to touch the sow between the ears for 10 s. An external observer registered the result as ‘0’ if the sow allowed the evaluator to touch her between the ears or ‘1’ if the sow did not allow the evaluator to touch her between the ears (adapted from [19]). Additionally, the vocalizations emitted by the sows during the test were evaluated as ‘0’ if there was no vocalization and ‘1’ if there was vocalization. At the beginning of the test, the first grunt was disregarded, and the following ones were in fact used to register that there was vocalization.

Saliva samples were collected in the mornings, five days after farrowing from a subset of sows (14 samples) for cortisol quantification. These sows were chosen randomly inside the blocks (parity order). Cotton ropes were utilized for saliva sampling, which were tied to the side of the farrowing crate at an accessible height for the sows to moisten the material for 30 min. After this time, through manual pressure on the rope, the saliva was collected and kept at −80 °C until analysis [20]. Cortisol levels were measured using commercial kits (Cortisol ELISA Kit, ELK Biotechnology, Wuhan, China) on a spectrophotometer (DR-200BN, Kasuaki, Stockholm, Sweden).

Blood samples were collected for serotonin quantification from the same animals after saliva sampling. The animals were individually restrained by nose snaring positioned behind the canine teeth and held in the correct position with the neck lifted upward to facilitate access to the vein. The collection was performed by puncture of the cranial vena cava using hypodermic needles and syringes with EDTA. Blood was centrifuged at 3000 rpm for 10 min, and plasma was separated and frozen at −20 °C until analysis. The quantitative determination of the plasma serotonin concentration was performed using an enzyme immunoassay (5-Hydroxytryptamine ELISA Kit, ELK Biotechnology, Wuhan, China) in a spectrophotometer (DR-200BN, Kasuaki, Stockholm, Sweden).

### 2.4. Piglet Assessments

At 14 days of age, a back test was conducted in all piglets in the litter. For the test, each animal was individually positioned in dorsal decumbency using a table positioned in front of each pen. The animal was observed for 60 s, during which the number of escape attempts, the latency to first vocalization, the number of vocalizations, and the frequency of vocalization throughout the test were registered [21,22]. Grunts, squeals, and screams were classified. Subsequent vocalizations had to be separated at least by 1 s of resting to be treated as two separate vocalizations. One person (evaluator 1) performed all the testing. During the test, evaluator 1 placed one hand on the piglet’s chest and the other hand supporting piglet’s hip. The four members were free. A second evaluator was responsible for timing the test and registering the data.

At the end of the experimental period (weaning), a total of 11 piglets were randomly selected (1 animal per sow), weighed, and euthanized. Immediately after, the abdominal cavity was opened, and the gut segments were isolated from each other. Spleen, thymus, heart, and adrenal glands were weighed, and the obtained value was relativized to the body weight of each piglet [23].

### 2.5. Statistical Analysis

Data were analyzed using SAS (v. 9.4, SAS Institute Inc., Cary, NC, USA). Responses were subjected to variance analysis using the GLIMMIX procedure. All statistical models included the effect of the treatment. Models with blocks (i.e., parity order) and their interactions were tested for each response and maintained in the final model when significant (*p* < 0.10). The binary responses (scores) were denoted through the utilization of a particular option. The residuals of the final analysis were stored and evaluated for normality using the Shapiro–Wilk test. The results were interpreted at the 5% (significant results) and 10% (tendencies) levels of significance.

## 3. Results

### 3.1. General Observations

The Sows and piglets performed as expected for modern genetics, with no health concerns detected during the study. The treatment effect on the performance was reported in a previous publication [24]. Briefly, no effects were observed on the total number of born alive, stillborn, and mummified piglets. However, the use of probiotics improved the alive birth weight (1.342 kg × 1.404 kg; *p* < 0.05) and weaning weight (5.725 kg × 5.329 kg; *p* < 0.001).

### 3.2. Sow Assessment

The sows supplemented with probiotics presented 33% lower HAR scores (*p* = 0.005; Table 1) or, in other words, were less reactive to the presence of humans. This effect was also observed in the afternoons, when the probiotic-supplemented sows exhibited a 31% better HAR score (*p* = 0.017). The probiotic still tended to improve (57%; *p* = 0.075) the HAR scores if only the results obtained in the mornings were considered. However, the results obtained in the morning presented greater variability (among sows) compared to the afternoon. There was no effect of the treatments on the vocalizations during the HAR test.

The frequency of stereotypies was not influenced by the treatments (Table 2). However, the sows supplemented with probiotics presented more standing positions (25%; *p* = 0.054) and less lying position (−15%; *p* = 0.008) in comparison to the control group. The frequency of the other positions and posture changes were similar between the treatments.

The sows treated with probiotics had a 50% lower concentration of salivary cortisol than the control sows at the end of gestation (*p* = 0.047; Table 3). In addition, the probiotics were able to improve the serotonin levels in the sows by 10% (*p* = 0.034).

### 3.3. Piglet Assessment

The piglets whose mothers were supplemented with probiotics tended to present the first vocalization earlier (*p* = 0.076), and they showed a lower time of vocalization throughout the test (*p* = 0.065) compared to the control group (Table 4). However, the number of vocalizations and the number of escape attempts were not influenced by the treatments. The piglet organ weights were also not influenced by the treatments (Table 5).

## 4. Discussion

Factors such as limited space within confinement and high productivity rates are inherent to intensified production systems, such as the one where the study was developed [25,26]. These factors can affect animals at the behavioral, emotional, and physiological levels. Increased stress levels emerge as one of the consequences, triggering disturbances in the body’s homeostasis and increasing abnormal behaviors [27]. The gut microbiota plays a role in brain function by regulating stress and cognition [28]. The mechanisms that connect intestinal bacteria and behavioral patterns are numerous, but not all of them are known or fully understood [11]. Through the production of metabolites, the regulation of the immune system, hormonal patterns, and the modulation of neurotransmitters, lactic acid bacteria are capable of modifying host brain functions and thereby shaping their behavior [29].

The HAR is well recognized as an effective test to assess the fear of animals towards humans [19,30]. A previous study showed that confident sows exhibited greater numbers of piglets born and weaned, indicating a positive association between sow behavior and reproductive performance [8]. In addition, sows that exhibit better HAR responses are also more likely to accept daily handling practices, which can minimize their susceptibility to injuries resulting from aversive or aggressive behavior towards humans [31]. Consequently, factors that may benefit the human–animal relationship hold significant importance both for the animal and the production chain.

The HAR assessment might be influenced by different periods of the day (morning and evening) because cortisol secretion follows a circadian rhythm, with the peak occurring in the morning followed by a decline during the day [32]. In the control group, a typical biological pattern (i.e., in accordance with the cortisol secretion pattern) was observed, with higher HAR values in the morning and a decrease in the afternoon. The probiotic group, on the other hand, exhibited a decrease in aversion and reactivity behaviors as well as a reduction in the circadian fluctuations of these behaviors.

Vocalization measures can also be used as biological markers of emotional reactivity, which is defined as the predisposition that an individual has in the face of a challenging situation [33]. However, in this study, vocalizations were not affected by the probiotic supplementation.

Stereotyped behavior has been widely recognized as a manifestation of frustration, apathy, or a lack of motivation [26]. Stereotypies are observed in sows especially in environments of intensive production, absence of stimuli, or restrictions on the animal’s freedom to engage in natural behaviors [34]. These behaviors have no clear purpose and can lead into injuries to the animals, further exacerbating their stress level [25].

In a previous study, the authors hypothesized that the expression of stereotypies might not be correlated with the increase in salivary cortisol concentrations during the first third of pregnancy [26]. This hypothesis suggested that stereotyped behaviors could be a form of relief, compensating for the absence of external stimuli. However, the relationship between these factors was not established, as it failed to persist throughout the remaining sections of pregnancy during subsequent assessments [26]. In the present investigation, no relationship was established between probiotics and stereotyped behaviors, despite the considerable impact of the additive on cortisol levels.

In this study, the supplemented sows spent more time standing and less time lying down, which may indicate that the probiotic acted to mitigate the depressive and apathetic state of the supplemented sows. Probiotics can modulate the intestinal microbiota, which has been linked to changes in behavioral patterns [10]. It is known that stressed sows change position more frequently [35]. No effects of the probiotics on position changes were found in this study. However, in confined conditions, sows take longer to transition from one position to another because of both intrinsic and extrinsic factors [36,37]. Thus, the likelihood of the treatment having an effect could be reduced by the fact that postural changes were also minimal in the control group.

Decreased motivation, anhedonia, and physical exercise are behavioral patterns commonly observed in humans and rats diagnosed with depression and anxiety disorders [38]. These patterns are attributed to the reduced sympathetic nerve function, which is involved in the unconscious mechanisms of warning signals and energy expenditure [39]. Studies that evaluated the action of *Lactobacillus casei* on the autonomic nervous system in rats subjected to stress observed a suppression of the afferent sympathetic nerve output to the adrenal gland and a suppression of the activation of the hypothalamic–pituitary–adrenal axis [40,41]. In the current study, the probiotic supplementation likely had an impact on the autonomic nervous system, leading to a reduction in symptoms associated with lethargy and demotivation. Consequently, the sows displayed increased activity, as evidenced by their greater tendency to stand and reduced inclination to lie down.

*Bifidobacterium* and *Lactobacillus* were able to reduce the levels of corticotropin-releasing hormone [42]. This hormone is the main stimulator of the release of adrenocorticotropic hormone, responsible for inducing the release of cortisol by the adrenal cortex [43]. The effects of probiotics on cortisol are particularly important for intensive pig production because long-term strict confinement can induce apathy and boredom in sows [44], leading them to expressing depressive behavior, often associated with increased cortisol levels [38]. Cortisol can cross the blood–brain barrier at the hypothalamic–pituitary–adrenal axis and can affect the regulation of the brain region that controls emotions [25].

It has already been elucidated that host–microbial imbalance is associated with mental disorders and stress-related illnesses in humans [45]. Probiotics play an important role in the regulation of the intestinal microbiota, thereby mitigating the effects of stress [2]. Certain strains, like *Lactobacillus* and *Bifidobacterium*, have the ability to influence brain functions [46] by participating in the communication axis between the brain and intestine through different pathsways, including the expression of neurotransmitters, dopamine, and serotonin [47]. Notably, the increase in plasmatic serotonin can be explained due to the synthesis of serotonin being dependent on several families of bacteria in the intestinal microbiota [48]. The microbiota can act on enterochromaffin cells present in the intestinal epithelium and specialized in the production of serotonin [49]. Being an important neurotransmitter for the communication of the microbiota–gut–brain axis, serotonin acts on the central nervous system through the serotonergic system, which is responsible for aspects such as stress response, emotional states, food impulse, and circadian rhythm, among others [50].

Typically, piglets exhibit a greater degree of activity and display a more extensive range of behaviors. The back test is one of the assessments that can be applied in this category and is widely used to evaluate fear, resistance, personality, stress responses, and coping style [21,22]. Longer and numerous vocalizations during the test are generally viewed as indicative of a proactiveness, also indicating a higher level of resistance [51]. The piglets from the supplemented sows showed less reactive behavior, which was characterized by less resistance, increased passivity, and milder responses.

Previous studies have associated the proactive behavioral profile, characterized by a higher vocalization percentage and a longer initial vocalization duration during the back test, with increased aggressiveness in animals introduced to a new social group [16,52,53]. The expression of aggressive behavior is related to the occurrence of fights, difficulty during handling, and elevated risk of injuries, which can depreciate the carcasses [54]. Other authors have also pointed out that a higher frequency and duration of vocalizations during piglet behavior tests can be an indication of stress, as the expression of certain types of vocalizations are affected by adrenaline discharges [28].

Bacteria resident in the gut affect the central nervous system through bidirectional communication via the microbiota–gut–brain axis. For newborn piglets, the maternal microbiota is one of the main sources for the establishment of their own microbial community [34,51]. Previous research has attested that the behavioral and physiological profiles of the sow can affect the profile of the progeny. For instance, sows that react with fear to humans tend to produce piglets that also react with fear [16]. Therefore, the current findings on the effects of the probiotics on piglet behavior could be explained by both the direct maternal influence on the piglet behavior (because the probiotics modulated the sow behavior) and also the great importance of the mother on the bacterial colonization of the offspring (because the probiotics modulated the sow gut microbiota), considering the transfer of the maternal microbiota to the piglets at the time of birth and during breastfeeding.

Furthermore, studies show that probiotics affect the neuroendocrine system [16]; therefore, supplemented sows could present calmer behavior and consequently a better maternal ability, which makes the stressful farrowing moment faster, which benefits the survival of piglets after birth as it reduces the number of uterine contractions and the consequent chance of hypoxia in intrauterine neonates [55]. Piglets that suffer less stress during birth search for food quickly [56]. In addition, regarding the relationship between the behavior of sows with piglets’ responses in the back test, it is understood that the link is based on a combination of genetic, epigenetic, and environmental effects [57,58].

It was also hypothesized that the weights of certain organs could be affected by stress due to the existing association between physiological changes, such as anxiety conditions, as previously demonstrated in rats supplemented with *Bifidobacterium longum* [23]. However, this condition was not found in the current trial.

Chronic stress in sows has emerged due to the challenging conditions of the traditional intensive production system, which worsen welfare status and, consequently, their production rates. The study of the potential of probiotic bacteria to modulate the neuroendocrine system allows us to better understand neurological disorders and their various consequences in order to mitigate them. In this context, probiotics can be the “key” to positively contribute to the quality of life of sows.

## 5. Conclusions

The use of a multistrain probiotic proved to be a beneficial nutritional strategy to enhance the welfare of pregnant sows. The supplemented sows showed better human–animal interactions, without interfering with the occurrence of stereotypies. The use of probiotics reduced the cortisol levels and increased serotonin levels in the sows. The piglets from the supplemented sows tended to vocalize less during the back test. These findings provide a theoretical basis for understanding maternal–progeny integration through a microbial link. More research is needed to fully evaluate the effects of probiotics on welfare in different environments and production systems (e.g., collective pens).

## Figures and Tables

**Table 1 animals-14-00847-t001:** Effect of multistrain probiotic supplementation on human–animal relationship (HAR) and vocalizations of gestating sows.

Variables	Treatments ^1^		*p*-Value ^2^
	Control	Probiotic	
Sows, number	76	71	-
HAR ^3^ score—morning	0.923 (0.221)	0.400 (0.178)	0.075
Vocalization HAR ^4^—morning	0.153 (0.095)	0.111 (0.081)	0.736
HAR score—afternoon	0.627 (0.056)	0.433 (0.058)	0.017
Vocalization HAR—afternoon	0.538 (0.035)	0.529 (0.036)	0.859
HAR score—morning + afternoon	0.645 (0.054)	0.430 (0.055)	0.005
Vocalization HAR—morning + afternoon	0.514 (0.034)	0.492 (0.035)	0.659

^1^ Least-squares means with standard errors in the parentheses. ^2^ Probability of treatment effects. ^3^ Score of 0 was attributed when the sow allowed the evaluator to touch her between the ears, and score of 1 when the sow did not allows the evaluator to touch her between the ears. ^4^ Score of 0 when there was no vocalization, while 1 was attributed when there was vocalization.

**Table 2 animals-14-00847-t002:** Effect of multistrain probiotic supplementation the on stereotypes and posture assessment of gestating sows.

Variables	Treatments ^1^		*p*-Value ^2^
	Control	Probiotic	
Sows, number	66	56	-
**Stereotypes** ^3^			
Sham chewing	0.798 (0.050)	0.771 (0.055)	0.716
Tongue rolling	0.127 (0.026)	0.173 (0.029)	0.245
Bar biting	0.127 (0.025)	0.135 (0.028)	0.831
Floor licking	0.116 (0.024)	0.122 (0.026)	0.873
**Posture** ^4^			
Standing	0.277 (0.024)	0.347 (0.026)	0.054
Lying	0.780 (0.023)	0.662 (0.025)	0.008
Lean forward	0.044 (0.011)	0.045 (0.012)	0.998
Sitting	0.034 (0.010)	0.051 (0.011)	0.284
Change in posture	1.137 (0.028)	1.102 (0.031)	0.408

^1^ Least-squares means with standard errors in the parentheses. ^2^ Probability of treatment effects. ^3^ Score of 0 was attributed to no observed stereotyped behavior, while 1 was attributed to observed stereotyped behavior. ^4^ Score of 0 was attributed to no change in posture, while 1 was attributed when a change in posture was observed.

**Table 3 animals-14-00847-t003:** Effect of multistrain probiotic supplementation on the hormonal concentration of gestating sows.

Variables	Treatments *		*p*-Value ^1^
	Control	Probiotic	
Sows, number	7	7	-
Salivary cortisol, mcg/dL	0.665 (0.084)	0.335 (0.094)	0.047
Blood serotonin, ng/dL	151.5 (3.456)	166.5 (4.340)	0.034

^1^ Probability of treatment effects. * Least-squares means with standard errors in the parentheses.

**Table 4 animals-14-00847-t004:** Effect of maternal probiotic supplementation during gestation and lactation, evaluated by the piglet back test.

Variables	Treatments ^1^		*p*-Value ^2^
	Control	Probiotic	
Piglets, number	71	90	-
Escape attempts, number	2.014 (0.161)	1.833 (0.143)	0.404
Time to first vocalization, sec	20.32 (2.662)	14.05 (2.292)	0.076
Vocalizations, number	1.971 (0.131)	2.211 (0.116)	0.175
Vocalizations, % of time	34.36 (4.478)	23.42 (3.824)	0.065

^1^ Least-squares means with standard errors in the parentheses. ^2^ Probability of treatment effects.

**Table 5 animals-14-00847-t005:** Effect of maternal probiotic supplementation during gestation and lactation on the relative piglet organ weights.

Variables	Treatments ^1^		*p*-Value ^2^
	Control	Probiotic	
Piglets, number	6	6	-
Thymus, % BW	0.211 (0.033)	0.216 (0.036)	0.911
Heart, % BW	0.565 (0.032)	0.609 (0.035)	0.386
Spleen, % BW	0.519 (0.238)	0.211 (0.261)	0.406
Average adrenal ^3^, % BW	0.141 (0.017)	0.163 (0.019)	0.426

^1^ Least-squares means with standard errors in the parentheses. ^2^ Probability of treatment effects. ^3^ Mean of the left and right adrenal glands.

## Data Availability

Data are contained within the article.

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
