# Peer review of "Effects of Multistrain Probiotic Supplementation on Sows’ Emotional and Cognitive States and Progeny Welfare"

_animals, 2024, doi:10.3390/ani14060847_

Round 1

Reviewer 1 Report

Comments and Suggestions for Authors

1. A total of 166 sows  were used in the trial. The treatments were randomly assigned to the experimental units within each block (parity order). In order to achieve the desired number of experimental units and considering the availability of animals in the farm (n = 300), this study was developed in four replicates.

It is not clear how the animals were assigned and how many replicates for each parameter. 

2 What is the dosage of probiotics supplemented to each sow?

A total of 11 piglets were randomly selected for assess the relative piglet organ weights.  Have you observed the growth performance of animals?

How to link the behavior of sows with the indicators of piglets?

5. The authors should explain in the introduction and discussion sections why lactic acid bacteria can improve pig behavior.

6 . Why set significant at p< 0.10?

7 . Please add the number of replicatesin (n) in each Table.

8. Describe the results using statistical methods rather than comparing the numerical values between two groups.

9. This study found the treatment effect of probiotics, but the mechanism involved is very limited, which is a limitation of this study and should be emphasized for discussion.

Author Response

Thank you very much for your contribution. Your review was very valuable in improving this study. Please see the attachment.

Reviewer 2 Report

Comments and Suggestions for Authors

Very interesting and important paper. Please try to improve the Materials and methods chapter, especially in case of the methods used in the behavioural part of the research. Please clarify how the animal behaviour was observed and analysed, as well as how the vocalizations were recorded. 

Comments on the Quality of English Language

Moreover, please try to avoid word "stereotypes" in plural.  Instead, please try to use stereotypies or behavioural stereotypies or stereotypic behaviour. In my opinion the paper is good enough to publish it in Animals after some minor changes and language corrections.

Author Response

(The authors gave the same response as above.)

Reviewer 3 Report

Comments and Suggestions for Authors

This is an interesting study on the effect of probiotic supplementation on the behaviour and welfare of sows during gestation and lactation and on their piglets, but the manuscript must be improved in all sections. All aspects that should be considered are detailed below:

Introduction

The introduction is not properly organized and does not fully address the issues discussed in the paper. 

Page 2, 2nd paragraph: studies 3 and 4 referenced do not use probiotics. Please use appropriate references.

Material and methods

In general, it is not clear how many samples were used in each study. There is a lack of justification that the number of samples used in each section is the statistically required number to be able to draw the conclusions indicated.

2.2. Experimental design, treatments and feeding practices.

How was the “n” required for the study calculated?

It is not clear how many animals were used in each group: 300 total, half for the control group?; four replicates? Please, indicate it more clearly.

Regarding the probiotic, in what proportion and quantity each microorganism is present in the product used? How much of each microorganism is administered to the animals in each feeding?

It is stated that "The capsule was placed on the top of the feed served to each sow in the morning. Red capsules were used to allow easy observation of its intake by the sow." Was it verified in all cases that the animals consumed the capsule? If so, please indicate.

2.3. Sow assessments

What do you mean by preliminary study? Is it not part of the work you are presenting?

The number of saliva samples was 14 for each group (control and probiotic)? Were there also 4 replicates? Please clarify. It is a very small number in relation to the total number of animals, how was the “n” needed for this study calculated? When were the samples taken, in the morning or in the afternoon?

2.4. Piglet assessments

How many piglets were back-tested? Were 11 piglets from each group (control and probiotic) euthanized? Please, clear it in the text.

2.5. Statistical analysis

Why was P<0.10 chosen and not P<0.05?

3. Results

Please, indicate the number of samples (n) in the tables.

Please, unify the way of indicating clarifications (superscripts or asterisk).

In table 4, what does the “n” mean?

4. Discussion

Please arrange the discussion of the different parameters investigated in the same order as they appear in the results.

This section should focus on discussing the results, interpreting them, comparing them with those of other authors and trying to justify them. It should not merely be a bibliographic review.

The third paragraph of the discussion states: "In the control group, a typical biological pattern (i.e., in accordance with the cortisol secretion pattern) was observed, with higher HAR values in the morning and a decrease in the afternoon"; has morning and afternoon cortisol been investigated in the study? Please make it clear in the text.

The serotonin results have not been discussed.

Page 8, last paragraph: the sentence "It is well established that gut bacteria affect the central nervous system through bidi-rectional communication via the microbiota-gut-brain axis" also appears in the introduction.

The conclusions should be nuanced, as many factors (including environmental factors) influence the animal welfare and quality of life of the sows. The limitations of the study should be included.

Author Response

Thank you very much for your contribution. Your review was very valuable in improving this study. Please see the attachment

Reviewer 4 Report

Comments and Suggestions for Authors

The author studies multistrain probiotics supplementation on sow’s behavior and progeny welfare.

Materials and Methods

“A total of 166 sows (Pic Camborough, Agroceres-PIC, São Paulo, Brazil)….”

I believe it should be PIC.

Please provide a summary of the parity distribution for each treatment.

Please provide detailed information on the housing environment, such as control of the cooling system for gestation and lactation.

 Were all sows housed in the same facilities?

How was culled sow data handled? Were sows from four replicates contained the sows in the first replicate?    

How many piglets per litter after cross-fostering?

2.2. Experimental design, treatments and feeding practices.

Was the initial BW of sows used as a factor for an allotment?

Disappearance of Capsule is unnecessary to indicate uptake only by sows and not spread on the feed.  

2.3. Sow assessments

How many saliva samples were collected in total?

Was sow behavior evaluated during lactation?

For behavior assessment, how is the order of each sow decided for each day? Which treatment was evaluated first? 

2.4. Piglet assessments

What criteria were used for piglet selection? Were piglets selected for sampling having similar BW and from the same sow parity? There were 166 sows used in this trial. Are 11 piglets enough to represent the whole population?

2.5. Statistical analysis

What design was used? What was the experimental unit?

3.1. Sows assessment

“This effect was also observed in the afternoons,”

This sentence required reworking. The HAR score is composed of morning and afternoon. Therefore, morning and afternoon observation lead to HAR scores.

What was the behavior score for those sows that were selected for blood cortisol and serotonin determination?

4. Discussion

A previous study showed that confident sows exhibited great numbers of piglets born and weaned, indicating a positive association between sow behavior and reproductive performance [8].”

Are reproductive performance data available for this study?

Author Response

(The authors gave the same response as above.)

Reviewer 5 Report

Comments and Suggestions for Authors

The authors investigated on "Effects of multistrain probiotic supplementation on sows' emotional and cognitive states and the progeny welfare".

The study conducted is interesting, however some considerations must be reported.

-Section: Intruduction

In the introduction I recommend including the role of cortisol and serotonin in evaluating the well-being of sows.

-Section: Materials and Methods

Please, enter the protocols used for determining cortisol and serotonin concentrations

- Section: Discussion

It would be appropriate to report the mechanism of action of probiotic bacteria on the reduction of cortisol levels and increase in serotonin. Only hints are made without delving into the mechanisms of action

-Section: Conclusion

The conclusions should enhance the results, please improve it

Author Response

(The authors gave the same response as above.)

Round 2

Reviewer 1 Report

Comments and Suggestions for Authors

No

Reviewer 3 Report

Comments and Suggestions for Authors

The revisions made are adequate.

Page 4, 4th paragraph: "at the beginning of the test" is repeated.

Reviewer 4 Report

Comments and Suggestions for Authors The author addressed my comments and this manuscript can be accepted in present form.

Comments on the Quality of English Language The author addressed my comments and this manuscript can be accepted in present form.